# Beyond Layers: A Global Message-Passing Mechanism for Heterophilic Graphs

## Abstract

The effectiveness of most graph neural networks is largely attributed to the message-passing mechanism. Despite the significant success in homophilic graphs (*i.e.*, similar nodes are connected by edges), message-passing mechanism in heterophilic graphs (*i.e.*, dissimilar nodes are connected by edges) is still challenging. Due to the existence of low-order but dissimilar neighbor nodes in a path, messages from similar but high-order neighbor nodes are often weakened. In this paper, firstly, we conduct both theoretical and empirical analysis of the layer-by-layer local nature of the message-passing mechanism. Then, we propose a novel GloMP-GNN for heterophilic graphs by comprehensively introducing global insights into the message-passing mechanism. 1) During the message propagation phase, the global insight is introduced from the perspective of graph structure. We design a structure-based global propagation strategy, where messages can be effectively propagated with the bridge of virtual edges between a global virtual node and graph nodes. Moreover, a global edge adaption approach is included to aggregate messages with adaptive edge weight adjustment. 2) During the feature updating phase, the global insight is introduced with a feature-augmented compensatory updating method. Through a multi-view feature updating mechanism, the node feature representation can be effectively augmented by compensating the weakened message from different views. Finally, we conduct extensive experimental evaluations on eight datasets, which demonstrate the superiority of our proposed GloMP-GNN. As broader impacts, GloMP-GNN consistently performs well across multiple layers and also effectively prevents the over-smoothing problem. Codes are available on Github[1].

## 1 Introduction

Graph Neural Network (GNN) has emerged as an important method for graph representation learning, which have been widely used across various fields, such as social network analysis Huang et al. (2024); Yang et al. (2022), bioinformatics Zhang et al. (2024); Liu et al. (2024), and financial risk assessment Wang et al. (2023); Qian et al. (2024). The effectiveness of most GNNs is largely attributed to the message-passing mechanism Gilmer et al. (2017a), a prevalent paradigm that aggregates information from neighbor nodes to update the representation of nodes. Despite the significant success in homophilic graphs (*i.e.*, similar nodes are connected by edges), message-passing mechanism in heterophilic graphs (*i.e.*, dissimilar nodes are connected by edges) is still challenging. Due to the existence of low-order but dissimilar neighbor nodes in a path, messages from similar but high-order neighbor nodes are often weakened.

To tackle the challenges posed by heterophilic graphs, several advanced methods have been developed to enhance the message-passing mechanism Zheng et al. (2022); Luan et al. (2024). Approaches like blending high-order neighbors Zhu et al. (2020); Song et al. (2023); Wang & Derr (2021) and identifying potential neighbor nodes Pei et al. (2019); Suresh et al. (2021) aim to expand the effective neighborhood, but they may also amplify intermediate layers, introducing noise and over-reliance on irrelevant information. Other strategies focus on optimizing message aggregation, such as adaptive message aggregation Yan et al. (2022), layer-specific weight learning Chien et al. (2020), and diverse aggregation schemes Luan et al. (2022); Maurya et al. (2022); Du et al. (2022). Additionally,

---

[1]https://github.com/Anonymous-GloMP-GNN/GloMP-GNN

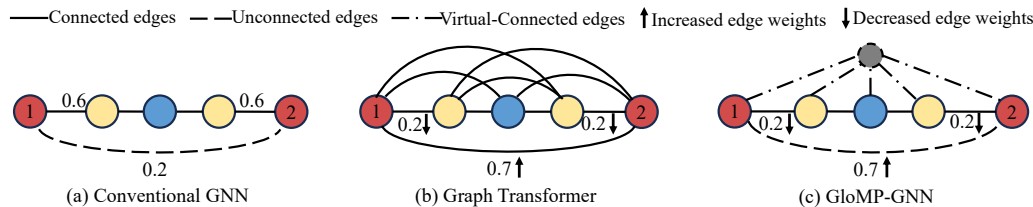

Figure 1: Illustration of different GNN methods in heterophilic graphs.

spectral methods differentiate between distinct-class neighbors using signed messages to capture high-frequency signals Yang et al. (2021); Bo et al. (2021). With continuous efforts, previous methods have alleviated the problem of the message-passing mechanism from different perspectives and achieved remarkable progress on heterophilic graphs.

However, few studies have fundamentally pointed out and solved the underlying problem, which is largely caused by the layer-by-layer localized nature of the current message-passing framework. Unlike these methods, we first theoretically and empirically analyze the localized layer-by-layer nature of the message-passing mechanism. Along this line, we propose to address the problem by introducing global insights into the message-passing mechanism. As illustrated in Figure 1, on heterophilic graphs, the message from similar but high-order neighbor nodes is often weakened by low-order dissimilar neighbor nodes in conventional GNN. Graph Transformer Shi et al. (2021) alleviates the issue by establishing global dependencies between nodes with fully-connected edges, but this will also greatly increase the extra quadratic computational complexity. Moreover, recent studies also reveal the over-globalizing problem Xing et al. (2024) in Graph Transformers with fully-connected edges. Unlike Graph Transformer, inspired by the concept of virtual node Gilmer et al. (2017b), we introduce global insights on heterophilic graphs by establishing virtual edges through a global virtual node, with only linear extra complexity.

To tackle the localized layer-by-layer nature of the message-passing mechanism, in this paper, we comprehensively introduce global insights into conventional message-passing mechanism and propose a novel Global Message-Passing Graph Neural Network (GloMP-GNN) for heterophilic graphs. To be concrete, the global insights of GloMP-GNN are reflected in two aspects. 1) During the message propagation phase, the global insight is introduced with a structure-based global propagation (SGP) strategy from the perspective of graph structure. By adding a global virtual node, messages between similar but high-order neighbor nodes can be effectively propagated with the bridge of virtual edges between the virtual node and graph nodes. Moreover, for redundant and noisy edges, a global edge adaption approach is included in SGP to adaptively aggregate messages by adjusting related edge weights; 2) During the feature updating phase, the global insight is introduced with a feature-augmented compensatory updating (FCU) method from the perspective of node feature. Through a multi-view feature updating mechanism, the node feature representation can be effectively augmented by compensating the weakened message from different views. The main contributions of our work are summarized as follows:

- We theoretically and empirically analyze the localized layer-by-layer nature of message-passing mechanisms. By comprehensively introducing global insights from both structure and feature perspectives, we propose GloMP-GNN with a global message-passing mechanism for heterophilic graphs.

- We propose a structure-based global propagation strategy by establishing virtual edges between the global virtual node and graph nodes. Moreover, a global edge adaption approach is included to aggregate messages with adaptive edge weight adjustment. In this way, messages between similar high-order neighbor nodes can be effectively propagated.

- We propose a feature-augmented compensatory updating method with multi-view feature updating mechanism. In this way, the node feature representation can be effectively augmented by compensating the weakened message from different views.

- Extensive experimental results on eight datasets demonstrate the superiority of our proposed GloMP-GNN. As broader impacts, GloMP-GNN consistently performs well across multiple layers and alleviates the over-smoothing issue.

## 2 PRELIMINARIES

### 2.1 BACKGROUND

Consider a graph $\mathcal{G} = (\mathcal{V}, \mathcal{E})$, where $\mathcal{V}$ represents the set of nodes and $\mathcal{E}$ denotes the set of edges. If nodes $i$ and $j$ are connected, then $(i, j)$ is an edge in $\mathcal{E}$. The adjacency matrix of $\mathcal{G}$ is represented by $\mathbf{A} \in \mathbb{R}^{N \times N}$, where $\mathbf{A}_{i,j} = 1$ if $(i, j) \in \mathcal{E}$ and $\mathbf{A}_{i,j} = 0$ otherwise. $N = |\mathcal{V}|$ indicates the number of nodes. The neighbor set of node $i$ is $\mathcal{N}(i) = \{j : (i, j) \in \mathcal{E}\}$. Each node $i \in \mathcal{V}$ has an associated $d$ dimensional feature vector $\mathbf{x}_i \in \mathbb{R}^d$ from the feature matrix $\mathbf{X} \in \mathbb{R}^{N \times d}$.

In the traditional GNN framework, the feature representation of each node is updated by aggregating information from its local neighbors. The process can be represented as:

$$\mathbf{m}_i^{(l)} = \text{AGG}^{(l)} \left( \left\{ \mathbf{x}_j^{(l-1)} : j \in \mathcal{N}(i) \right\} \right), \quad \mathbf{x}_i^{(l)} = \text{UPDATE}^{(l)} \left( \mathbf{x}_i^{(l-1)}, \mathbf{m}_i^{(l)} \right), \quad (1)$$

where $\mathbf{m}_i^{(l)}$ and $\mathbf{x}_i^{(l)}$ are the message vector and the feature representation of node $i$ at layer $l$, respectively. The function $\text{AGG}^{(l)}$ and $\text{UPDATE}^{(l)}$ are the aggregation function and update function.

### 2.2 ANALYSIS ON MESSAGE-PASSING MECHANISM

The classic matrix representation for message-passing GNNs, like GCN and GAT Kipf & Welling (2016); Veličković et al. (2018), can be written as $\mathbf{X}^{(l)} = \sigma(\hat{\mathbf{A}}^{(l)} \mathbf{X}^{(l-1)} \mathbf{W}^{(l)})$. In GCN, $\hat{\mathbf{A}}^{(l)} = (\mathbf{D}+\mathbf{I})^{-1/2}(\mathbf{A}+\mathbf{D})(\mathbf{D}+\mathbf{I})^{-1/2}$, where $\mathbf{D} = diag(\mathbf{A})$ is a diagonal matrix. In GAT, $\hat{\mathbf{A}}^{(l)} = \mathbf{A} \circ \mathbf{M}^{(l)}$, where $\circ$ represents the element-wise multiplication, $\mathbf{M}^{(l)}$ represents the attention coefficient matrix at layer $l$. To simplify the mathematical exploration of model properties, following Eliasof et al. (2023); Azabou et al. (2023), the $\sigma(\cdot)$ function (*i.e.*, *ReLU*) is omitted in the following parts. Then, for traditional GNN models, $\mathbf{X}^{(l)} = \prod_{i=1}^{l} \hat{\mathbf{A}}^{(l-i+1)} \mathbf{X}^{(0)} \mathbf{W}^{(i)}$. Derivation details are listed in Appendix A.1.

In Graph Neural Networks (GNNs), the traditional qualitative descriptors of node relationships, such as similarity or strength, are insufficient for a detailed analysis of node interactions. We introduce influence intensity as a quantitative metric to overcome these limitations by measuring the exact degree of influence between nodes, accommodating both direct and indirect interactions. Specifically, we define the "*Global Influence Intensity*" and "*Path Influence Intensity*" as follows.

**Definition 1 (Global Influence Intensity)** *In an $l$-layer GNN, the global influence intensity from node $q$ to node $p$, denoted as $C_{p,q}$, is calculated by the matrix element $(\prod_{i=1}^{l} \hat{\mathbf{A}}^{(l-i+1)})_{p,q}$.*

**Definition 2 (Path Influence Intensity)** *The influence intensity of node $q$ on node $p$ along a specific path $P = (p, i_1, i_2, \ldots, i_{k-1}, q)$ is denoted as $C_{p,q}^P$, which is computed based on the weights along path $P$.*

Obviously, the Global Influence Intensity from node $q$ to node $p$ can be calculated by adding all the Path Influence Intensity of node $q$ on node $p$. Under the definition of influence intensity, it's obvious that for a specific path $P = (i, j)$, $C_{i,j}^P = \hat{A}_{i,j}$, where edge weight and influence intensity between node $i$ and $j$ are numerically identical.

Considering that there may be multiple paths between two nodes in a graph, the two nodes may be different order neighbors on different paths. In order to unify the description, we define the neighbor orders of nodes as follows.

**Definition 3 (k-order Neighbors)** *The $k$-order neighbors of a node $p$ in a graph encompass all neighbors that are at a minimum hop of $k$ from the node $p$, denoted as $\mathcal{N}^{(k)}(p)$.*

Along with the previous introduction, we intend to investigate the propagation of the influence intensity in the message-passing mechanism. Taking the widely adopted GCN and GAT as illustrative examples, we perform an in-depth analysis of 10-layer GNNs.

From Figure 2(a), we observe a consistent trend for both GCN and GAT. As the order of neighbors increases, the averaged global influence intensity begins to decrease. From Figure 2(b), we can

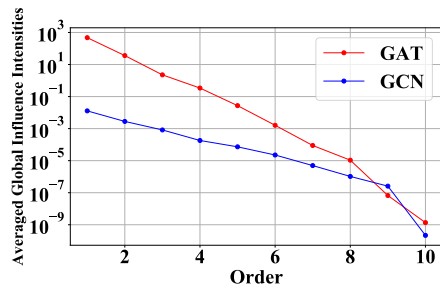 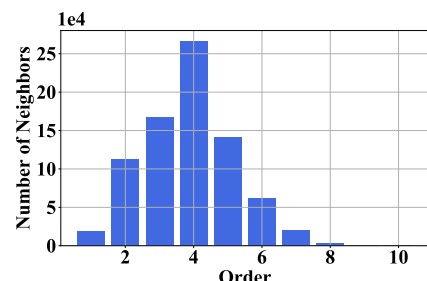

(a) Averaged global influence intensities from different orders of neighbors for GCN and GAT.

(b) Total numbers of neighbors for each order in the dataset.

Figure 2: Propagation of node influence intensity for GCN and GAT (10 layers) in the filtered Chameleon dataset Platonov et al. (2023).

observe that in a heterophilic graph, the number of nodes with high-order neighbors (for example, greater than 3) actually accounts for a very large proportion. However, the trend in Figure 2(a) implies that the influence of these high-order neighbors on the central node is extremely low. This tendency may be detrimental to capturing the broader structure in heterophilic graphs, where insights from higher-order neighbors are essential Zhu et al. (2020); Song et al. (2023).

The above illustration uses empirical analysis to show the global influence intensity between nodes in a graph under the layer-by-layer nature. Building upon these empirical findings, we now proceed with a theoretical examination to further demonstrate how the path influence intensity between nodes may be diminished by the inherent nature of the layer-by-layer message-passing mechanism.

**Theorem 1** *In traditional GNN models, for any given node $i_0$ and its $k$-order neighbor $i_k$ along any path $\mathcal{P} = (i_0, i_1, i_2, i_3, \ldots, i_k)$ within a heterophilic graph $\mathcal{G} = (\mathcal{V}, \mathcal{E})$, the path influence intensity of $i_k$ to $i_0$ (i.e., $C^{\mathcal{P}}_{i_0, i_k}$) approaches zero as $k$ tends towards infinity, i.e.,*

$$\lim_{k \to \infty} C^{\mathcal{P}}_{i_0, i_k} = 0.$$

The proof of the theorem is deferred to Appendix A.1. This theorem highlights the intrinsic limitation of the layer-by-layer nature of the message-passing mechanism, particularly in heterophilic graphs. In such graphs, nodes that are close yet dissimilar might receive lower influence weights, potentially weakening the contribution of distant but similar nodes in the same path. Although recent models like JKNet Xu et al. (2018), GPRGNN Chien et al. (2020) and GCNII Chen et al. (2020) have introduced residual connections to preserve self-information and jumping links or learning weights separately for each layer, the influence intensities of distant nodes can hardly benefit from this. Since an increase in the weight of a distant layer can affect the weights of intermediate layers, this may introduce noise and an overreliance on intermediate layers that may contain irrelevant or even harmful data. As a result, the influence intensities from high-order neighbors are very small, presenting a challenge in capturing valuable information from distant nodes for heterophilic graphs.

## 3 GLoMP-GNN

In this section, we will present our proposed GloMP-GNN with a global message-passing mechanism for heterophilic graphs in detail. Overall, GloMP-GNN comprehensively introduces global insights from two aspects. (1) From the perspective of graph structure, a structure-based global propagation strategy is designed in the message propagation phase. (2) From the perspective of node feature, a feature-augmented compensatory updating method is developed in the feature updating phase. The technical details of GloMP-GNN are presented as follows.

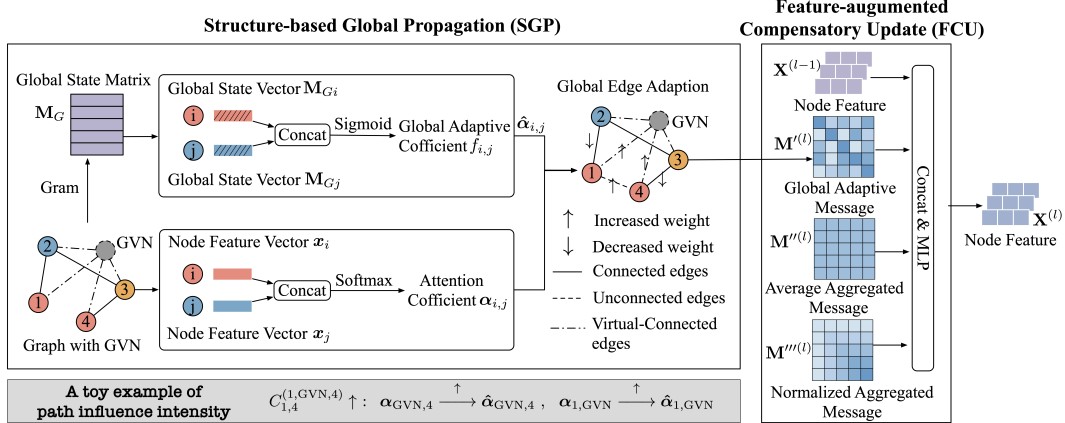

Figure 3: The framework of GloMP-GNN, which consists of a Structure-based Global Propagation module and a Feature-augmented Compensatory Update module. A toy example of path influence intensity illustrates that, with GVN and Global Edge Adaptation in SGP, the path influence intensity from node 4 to node 1 can be increased.

### 3.1 Structure-based Global Propagation

Inspired by the concept of virtual node Gilmer et al. (2017b); Cai et al. (2023), we add a Global Virtual Node (GVN) to connect every node in the graph. Then, virtual-connected edges between the GVN and graph nodes can be established. In this way, messages between similar but high-order neighbor nodes can be effectively propagated with the bridge of virtual edges. Therefore, the following theorem can be formulated:

**Theorem 2** *In graph $\mathcal{G}$, if a global virtual node is added to connect every node on the graph, then for any node on the graph, its maximum neighbor order is 2.*

Based on Theorem 2, messages can be effectively and efficiently propagated between similar but high-order neighbor nodes in heterophilic graphs.

For technical details, the feature representation of GVN (*i.e.*, $\boldsymbol{x}_{\text{GVN}}$) is initialized as:

$$x_{\text{GVN},i} = \max_{u \in \mathcal{V}} x_{u,i}, \tag{2}$$

where $x_{\text{GVN},i}$ is the feature representation of the GVN for the $i^{th}$ dimension. $\mathcal{V}$ represents all nodes in the graph, $x_{u,i}$ is the feature representation of node $u$ for the $i^{th}$ dimension.

Despite the effectiveness of the virtual node in information propagation, it may also introduce some redundant and noisy edges, which interfere with the learning process and degrade model performance Hu et al. (2020; 2021).

Along this line, we further design a Global Edge Adaption (GEA) approach to adaptively adjust the weights of those edges. Specifically, we firstly leverage the multi-head attention mechanism to learn different weights of neighbors, which can be calculated as follows:

$$\boldsymbol{\alpha}_{ij} = \text{Softmax}(\text{LeakyReLU}\left(\mathbf{a}^T[\mathbf{W}_A \boldsymbol{x}_i \| \mathbf{W}_A \boldsymbol{x}_j]\right)). \tag{3}$$

Here, $[\cdot \| \cdot]$ denotes concatenation, $\mathbf{a}$ is a learnable weight parameter vector, and $\mathbf{W}_A$ is a learnable weight parameter matrix.

Next, we introduce a global state matrix $\mathbf{M}_G$, which leverages the Gram matrix Sreeram & Agathoklis (1994) to encapsulate the relationships of all nodes based on their initial feature representations. Here, the Gram matrix is defined as $Gram = \mathbf{X}^{(0)}\mathbf{X}^{(0)^T}$, where $\mathbf{X}^{(0)}$ represents the initial feature matrix of the nodes. Moreover, we incorporate a noise matrix (*i.e.*, $\mathbf{M}_{\text{noise}}$) into $\mathbf{M}_G$, which serves as a proxy for potential uncertainties and variances inherent in real-world datasets. Thus, the generalization

ability and robustness of our proposed method could be improved. The formulation is written as follows:

$$\mathbf{M}_G = Gram \cdot \mathbf{W}_G + \epsilon \cdot \mathbf{M}_{\text{noise}}, \tag{4}$$

where $\mathbf{W}_G$ and $\epsilon$ are trainable weight parameters. $\mathbf{M}_{\text{noise}} \sim \mathcal{N}(0, 1)$ is sampled from the standard normal distribution.

Based on the global state matrix $\mathbf{M}_G$, we calculated the global adaptive coefficient $f_{i,j}$ for connected nodes $i$ and $j$ based on their global state representations as follows:

$$f_{i,j} = \sigma([\mathbf{M}_{Gi} \| \mathbf{M}_{Gj}] \cdot \mathbf{W}_f + \mathbf{b}_f), \tag{5}$$

where $[\mathbf{M}_{Gi} \| \mathbf{M}_{Gj}]$ denotes the concatenation of the global state vectors of nodes $i$ and $j$, $\mathbf{W}_f$ and $\mathbf{b}_f$ are learnable weight matrix and bias term. $\sigma(\cdot)$ is the sigmoid activation function that ensures the global adaptive coefficients are constrained in the range $[0, 1]$.

Then, a new global multi-head attention coefficient $\hat{\boldsymbol{\alpha}}_{i,j}$ can be obtained by integrating the global adaptive coefficient $f_{i,j}$ with the original multi-head attention coefficient $\boldsymbol{\alpha}_{i,j}$. This process is formulated as follows:

$$\hat{\boldsymbol{\alpha}}_{i,j} = \text{Softmax}(\beta(f_{i,j} \cdot \boldsymbol{\alpha}_{i,j}) + (1 - \beta)\boldsymbol{\alpha}_{i,j}), \tag{6}$$

where $\beta \in [0, 1]$ is a trainable parameter that balances the influence of the adaptively adjusted attention coefficient and original attention coefficient. This formulation can enhance our model ability to capture global dependencies between nodes.

Finally, based on the global attention coefficients $\hat{\boldsymbol{\alpha}}_{ij}$, the message is propagated and aggregated.

$$\mathbf{m}_i^{\prime(l)} = \sigma\left(\sum_{j \in \mathcal{N}(i)} \hat{\boldsymbol{\alpha}}_{i,j}^{(l-1)} \mathbf{W}^{\prime(l-1)} \boldsymbol{x}_j^{(l-1)}\right), \tag{7}$$

where $\mathbf{m}_i^{\prime(l)}$ is the aggregated message at $l^{th}$ layer for node $i$ after aggregating information from its neighbors at $(l-1)^{th}$ layer, $\mathbf{W}^{\prime(l-1)}$ is a learnable weight matrix. Additionally, $\mathbf{M}^{\prime(l)} = \{\mathbf{m}_i^{\prime(l)}\}_{i=0}^N$ represent the global adaptive message matrix at $l^{th}$ layer .

## 3.2 FEATURE-AUGMENTED COMPENSATORY UPDATE

As stated before, in heterophilic graphs, messages from similar but high-order neighbor nodes are often weakened due to the local nature of the layer-by-layer message-passing mechanism. To this end, during the message propagation phase, as presented before, we design the SGP strategy to introduce global insight from the perspective of graph structure. In addition, during the feature updating phase, we also develop a feature-augmented compensatory updating method to introduce global insight from the perspective of node feature. Specifically, we comprehensively utilize three different message aggregation mechanisms for multi-view feature updating. In this way, messages weakened in a single view can be mutually compensated by messages from other views.

The first is the aggregation mechanism from GEA, which provides an attention-based global adaptive view. The second is the average aggregation mechanism, which provides an edge-balanced view. It is formulated as follows:

$$\mathbf{m}_i^{\prime\prime(l)} = \sigma\left(\mathbf{W}^{\prime\prime(l-1)} \frac{\sum_{j \in \mathcal{N}(i)} \boldsymbol{x}_j^{(l-1)}}{|\mathcal{N}(i)|}\right). \tag{8}$$

Thus, the average aggregated message matrix at $l^{th}$ layer can be written as $\mathbf{M}^{\prime\prime(l)} = \{\mathbf{m}_i^{\prime\prime(l)}\}_{i=0}^N$.

The third is the normalized aggregation mechanism, which provides a node degree-based view. It can be represented as follows:

$$\mathbf{m}_i^{\prime\prime\prime(l)} = \sigma\left(\mathbf{W}^{\prime\prime\prime(l-1)} \sum_{j \in \mathcal{N}(i)} \frac{\boldsymbol{x}_j^{(l-1)}}{\sqrt{\text{degree}(i) \times \text{degree}(j)}}\right). \tag{9}$$

Then, the normalized aggregated message matrix at $l^{th}$ layer can be written as $\mathbf{M}'''^{(l)} = \{\mathbf{m}_i'''^{(l)}\}_{i=0}^N$.

By bringing multi-view messages all together, the representation of node $i$ will be updated with a fusion of its own features and aggregated messages using a multi-layer perceptron (MLP) with two hidden layers and Gelu activation. It is represented as follows:

$$\boldsymbol{x}_i^{(l)} = \text{MLP}([\boldsymbol{x}_i^{(l-1)}\|\mathbf{m}_i'^{(l)}\|\mathbf{m}_i''^{(l)}\|\mathbf{m}_i'''^{(l)}]), \tag{10}$$

where $[\cdot\|\cdot]$ denotes the concatenation opration.

In this way, our proposed FCU method comprehensively utilizes messages in different views and augments the node feature updating process. Thus, FCU is capable of overcoming the local nature of current message-passing mechanisms from the perspective of node features.

## 4 EXPERIMENTS

### 4.1 DATASETS

To investigate the performance of our model across various datasets, we conduct experiments on five heterophilic datasets (Actor Pei et al. (2019), Roman-empire Lhoest et al. (2021); Platonov et al. (2023), Amazon-ratings Leskovec & Krevl (2014); Platonov et al. (2023), Minesweeper Platonov et al. (2023), and Tolokers Platonov et al. (2023)), and three commonly used homophilic datasets (Cora, CiteSeer, PubMed Sen et al. (2008)). For a more comprehensive measurement of dataset heterophily, we use two metrics: $h_{\text{edge}}$ Pei et al. (2019) and label informativeness (LI) Platonov et al. (2024). More descriptions of these datasets and heterophily metrics are listed in the Appendix A.2.

### 4.2 BASELINES

To verify the effectiveness of the proposed GloMP-GNN on the node classification task, 18 methods are employed as baselines, which can be divided into five groups: (1) deep learning method ResNet He et al. (2016); (2) classic GNN models, such as GCN Kipf & Welling (2016), GraphSage Hamilton et al. (2017), GAT Veličković et al. (2018) and GATv2 Brody et al. (2022); (3) selective information propagation method , such as H$_2$GCN Zhu et al. (2020), GBK-GNN Du et al. (2022), GCNII Chen et al. (2020), FSGNN Maurya et al. (2022), and OrderedGNN Song et al. (2023); (4) graph signal-based methods, such as GPR-GNN Chien et al. (2020), FAGCN Bo et al. (2021), JacobiConv Wang & Zhang (2022) and ALT-APPNP Xu et al. (2023); (5) global information-based method, such as Graph Transformer (GT) Shi et al. (2021), GraphGPS Rampášek et al. (2022), GloGNN++ Li et al. (2022) LRGNN Liang et al. (2024). More descriptions and analysis of these baselines are listed in the Appendix A.2.

### 4.3 EXPERIMENTAL SETUP

All experiments are implemented with PyTorch Paszke et al. (2019) and DGL Wang et al. (2019) on a Linux server equipped with six 2.30GHz Intel (R) Xeon (R) Gold 5218 CPUs and an NVIDIA Tesla V100-SXM2-32GB GPU. All models are trained with the Adam optimizer Kingma & Ba (2015). For GloMP-GNN, we explore a range of hyperparameters: learning rates are chosen from $\{1e-2, 1e-3, 1e-4, 3e-4, 1e-5, 3e-5\}$, hidden dimensions are taken from $\{64, 128, 256, 512\}$, the number of attention heads is set to 4 or 8, and the number of hidden layers varies from 1 to 10. For baselines, the experimental parameter settings are based on the hyperparameters provided in original papers, datasets, and our computational resources. All the models are tuned to be optimal to ensure fair comparisons. Models are trained for 1,000 epochs on ten 50%/25%/25% train/validation/test splits in heterophilic datasets and ten 60%/20%/20% train/validation/test splits in homophilic datasets. We select models based on the best validation set performance. Following Platonov et al. (2023); Müller et al. (2024), for a fair comparison, we also adopt Accuracy as evaluation metrics on Actor, Roman, Amazon, Cora, Citeseer and Pubmed datasets, and adopt AUC as metrics on Minesweeper and Tolokers datasets.

Table 1: Performance of GloMP-GNN and other popular GNN models on both heterophily and homophily graph datasets.

| Model | Actor | Roman | Amazon | Minesweeper | Tolokers | Cora | Citeseer | Pubmed |
|---|---|---|---|---|---|---|---|---|
| ResNet | 33.47±0.75 | 65.88±0.38 | 45.90±0.52 | 50.89±1.39 | 72.95±1.06 | 74.95±2.09 | 72.90±1.70 | 86.78±0.38 |
| GCN | 34.96±1.10 | 73.69±0.74 | 48.70±0.63 | 89.75±0.52 | 83.64±0.67 | 86.60±0.95 | 75.88±1.52 | 88.18±0.50 |
| GraphSage | 35.68±0.72 | 85.74±0.67 | 53.63±0.39 | 93.51±0.57 | 82.43±0.44 | 86.66±1.42 | 76.29±1.88 | 88.83±0.50 |
| GAT | 34.82±1.17 | 80.87±0.30 | 49.09±0.63 | 92.01±0.68 | 83.70±0.47 | 86.80±1.02 | 75.93±1.38 | 87.82±0.43 |
| GATv2 | 35.66±0.70 | 85.69±0.57 | 49.71±0.68 | 91.53±0.66 | 82.93±0.62 | 86.73±1.15 | 75.86±1.73 | 87.81±0.52 |
| H2GCN | 35.09±1.00 | 60.11±0.52 | 36.47±0.23 | 89.71±0.31 | 73.35±0.01 | 87.12±0.81 | 77.04±1.15 | 88.53±0.42 |
| GBK-GNN | 34.38±0.67 | 74.57±0.47 | 45.98±0.71 | 90.85±0.58 | 81.01±0.67 | 86.74±0.74 | 76.15±2.02 | 88.79±0.53 |
| GCNII | 34.88±0.85 | 79.33±0.56 | 49.70±0.68 | 89.64±1.18 | 84.89±0.54 | 86.12±0.88 | 76.24±1.83 | 88.80±0.43 |
| FSGNN | 35.21±0.66 | 79.92±0.56 | 52.74±0.83 | 90.08±0.70 | 82.76±0.61 | 85.49±1.15 | 75.65±1.42 | 89.31±0.37 |
| OrderedGNN | 36.01±1.13 | 81.92±0.79 | 52.35±0.55 | 90.13±1.77 | 81.85±0.87 | 86.96±1.44 | 75.48±1.73 | 89.07±0.52 |
| GPR-GNN | 34.70±0.86 | 64.85±0.27 | 44.88±0.34 | 86.24±0.61 | 72.94±0.97 | 87.63±1.59 | 77.15±1.67 | 88.58±0.48 |
| FAGCN | 34.95±1.36 | 65.22±0.56 | 44.12±0.30 | 88.17±0.73 | 77.75±1.05 | 87.89±0.85 | 76.35±1.12 | 89.32±0.28 |
| JacobiConv | 35.54±0.85 | 71.14±0.42 | 43.55±0.48 | 89.66±0.40 | 68.66±0.65 | 86.76±0.98 | 76.42±1.36 | 89.02±0.39 |
| ALT-APPNP | 32.41±1.27 | 69.13±0.43 | 43.81±0.37 | 80.19±0.26 | 78.60±0.62 | 85.01±0.86 | 73.54±0.60 | 89.06±0.48 |
| GT | 33.86±1.04 | 86.51±0.73 | 51.17±0.66 | 91.85±0.76 | 83.23±0.64 | 86.76±1.30 | 75.80±1.53 | 87.17±0.58 |
| GraphGPS | 36.53±0.68 | 87.04±0.58 | 51.03±0.60 | 93.85±0.41 | 84.81±0.86 | 86.56±1.01 | 76.02±1.17 | 88.94±0.57 |
| GloGNN++ | 35.42±0.76 | 59.63±0.69 | 36.89±0.14 | 51.08±1.23 | 73.39±1.17 | **88.33±1.09** | **77.22±1.78** | 89.24±0.39 |
| LRGNN | 36.86±0.86 | 62.29±1.33 | 36.79±0.49 | 80.00±0.00 | 78.51±0.38 | 88.26±1.02 | 75.19±1.51 | 89.26±0.62 |
| GloMP-GNN | **37.04±0.80** | **90.21±0.62** | **53.72±0.41** | **96.32±0.42** | **85.11±0.64** | 87.53±1.32 | 76.87±1.12 | **89.72±0.37** |

Table 2: Ablation Performance (%) of GloMP-GNN on different datasets.

| Model | Actor | Roman | Amazon | Minesweeper | Tolokers | Cora | Citeseer | Pubmed |
|---|---|---|---|---|---|---|---|---|
| (1) w/o GVN | 36.12 | 89.51 | 52.71 | 95.05 | 84.51 | 86.57 | 75.84 | 88.80 |
| (2) w/o GEA | 36.80 | 89.47 | 52.82 | 95.11 | 84.14 | 86.46 | 75.21 | 89.12 |
| (3) w/o $\mathbf{m}'$ | 36.87 | 86.82 | 52.72 | 94.62 | 84.06 | 86.31 | 75.72 | 89.48 |
| (4) w/o $\mathbf{m}''$ | 36.64 | 88.53 | 53.03 | 95.40 | 83.74 | 85.71 | 75.47 | 88.95 |
| (5) w/o $\mathbf{m}'''$ | 36.43 | 88.33 | 52.12 | 95.67 | 83.60 | 86.92 | 75.32 | 88.84 |
| (6) GloMP-GNN | **37.04** | **90.21** | **53.72** | **96.32** | **85.11** | **87.53** | **76.87** | **89.72** |

## 4.4 PERFORMANCES ON DIFFERENT DATASETS

We evaluate different methods on the aforementioned 8 datasets, following the same data splits as Pei et al. (2019); Platonov et al. (2023). As shown in Table 1, we report the average performance with the standard deviation on test sets over 10 data splits.

Compared with various state-of-the-art models across homophilic datasets and heterophilic datasets, GloMP-GNN is the most reliable and effective model across a wide range of datasets, demonstrating its superiority and robustness. Furthermore, it is observed that traditional GNN methods tend to outperform methods designed for heterophily on many datasets. This aligns with the issues identified in Platonov et al. (2023), indicating that current heterophilic graph models still have significant room for improvement. In Tolokers, edges are relatively dense, and the excellent performance of our model demonstrates the ability of the Global Edge Adaption (GEA) module to adjust edge weights adaptively. Conversely, in Roman-empire, which has sparser edges, the models that perform well are those that incorporate global context by allowing nodes to attend to information from distant parts of the graph, rather than just their immediate neighbors, such as GT and our GloMP-GNN. This highlights the exceptional capability of our GloMP-GNN in capturing a global perspective and effectively learning information from distant nodes. These observations from the two datasets demonstrate the effectiveness of GloMP-GNN in both sparse and dense datasets. In addition, we also perform time analysis to show the efficiency of GloMP-GNN in Appendix A.4.

## 4.5 ABLATION STUDY

To investigate the effectiveness of different components in GloMP-GNN, we further conduct ablation studies on different variants of our proposed GloMP-GNN. In the message propagation phase, as shown in Table 2 (1)-(2), when removing GVN and GEA separately, the performance of GloMP-GNN decreased with different degrees, which demonstrates both GVN and GEA are critical for introducing global properties in the message-passing mechanism and improve model performance.

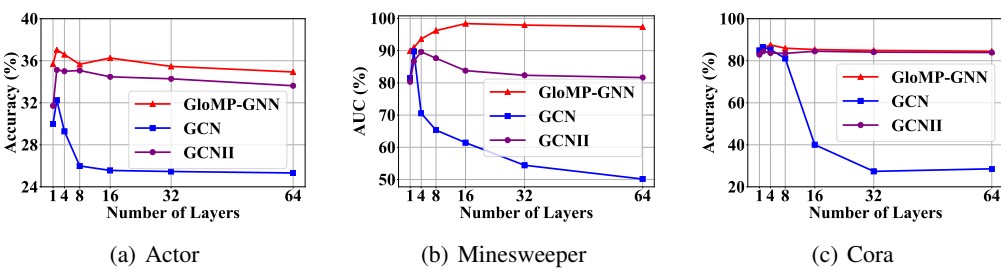

Figure 4: Node classification performances for over-smoothing problems with various depths.

In the feature updating phase, GloMP-GNN updates node features by comprehensively taking into account messages (*i.e.*, $\mathbf{m}'$, $\mathbf{m}''$ and $\mathbf{m}'''$) from different views. As shown in Table 2 (3)-(5), the performance of GloMP-GNN decreases with varying degrees when removing any message, which again illustrates the importance and benefits of our proposed multi-view feature updating mechanism.

### 4.6 OVER-SMOOTHING PROBLEM

To validate the ability of GloMP-GNN to mitigate over-smoothing, we compare its performance with GCN and GCNII on three datasets: Actor, Minesweeper, and Cora. In particular, GCNII is an approach specially designed to alleviate the over-smoothing issue. As shown in Figure 4, on the heterophilic Actor dataset, GloMP-GNN shows consistent performance across all layers, whereas GCN fluctuates, and although GCNII demonstrates moderate stability, its performance is not as strong as GloMP-GNN. On the heterophilic Minesweeper dataset, GloMP-GNN improves with increasing depth, while both GCN and GCNII decline, suggesting that GloMP-GNN effectively adapts to deeper networks. On the homophilic Cora dataset, GloMP-GNN peaks at the $4^{th}$ layer and maintains high accuracy with only a slight decline at 64 layers, while GCN exhibits significant drops in performance as depth increases. In summary, GloMP-GNN consistently alleviates over-smoothing, demonstrating its effectiveness across various graph structures and evaluation metrics. In addition, we also quantify the ability of GloMP-GNN to mitigate over-smoothing through Dirichlet Energy Karhadkar et al. (2023), which is shown in Appendix A.5.

### 4.7 CASE STUDY AND VISUALIZATION

**Visualization of Node Features.** On heterophilic graphs, node features learned in multi-layer GNNs are prone to over-smoothing. To investigate the ability of GloMP-GNN to solve the over-smoothing problem, we further conduct experiments on a heterophilic dataset (*i.e.*, Roman-empire). Specifically, we used t-SNE Van der Maaten & Hinton (2008) to visualize node representations learned on the 64-layer GCN and GloMP-GNN. The results are shown in the Figure 5. It is obvious that compared to GCN, node representations learned by GloMP-GNN are more discriminative. That is, intra-class node representations are close together, while inter-class node representations are far apart. The visualization results illustrate the potential ability of GloMP-GNN to alleviate the over-smoothing problem in multi-layer graph neural networks on heterophilic graphs.

**Case Study on Global Edge Adaption.** We also conduct a case study on the edge weight adjustment process by the Global Edge Adaption (GEA). For the convenience of observation and display, we have set the number of attention heads to 1 here. Taking the Roman-empire dataset as an example, as illustrated in Figure 6, each circle represents a node, with the corresponding ID below it. In the figure, the edge coefficient formed by nodes with node 1 and 2 is been reduced to a value close to 0, even though the original attention coefficient from node 2 to node 1 is relatively high at 0.6469, likely due to the similarity between their features, as computed in Equation 3. To analyze the reason, we investigated the 1-order and 2-order neighbors of node 2 and found that none of these neighbors belong to the same label as node 1. Therefore, the contribution of node 2 to node 1 is minimal. This demonstrates that GEA is capable of incorporating global information, effectively removing redundant and detrimental edges, thereby enhancing the accuracy of model learning.

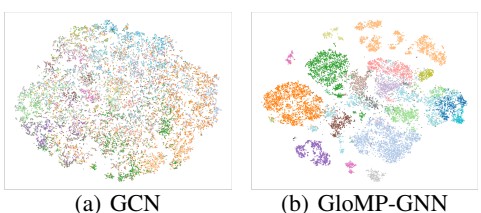

(a) GCN      (b) GloMP-GNN

Figure 5: Comparison of GCN and GloMP-GNN visualizations obtained from 64 layers of feature representations in two-dimensional space on Roman-empire dataset.

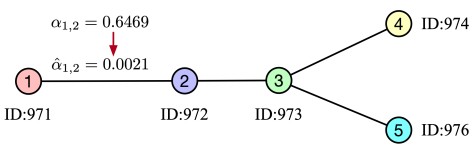

Figure 6: Example of edge weight adjustment by GEA within the Roman Empire dataset, where different colors represent different node labels.

## 5 RELATED WORK

In heterophilic graph neural networks, various strategies have been proposed to enhance effectiveness Zheng et al. (2022); Zhu et al. (2023); Khoshraftar & An (2024). A central approach is ego- and neighbor-embedding separation, effectively employed by models such as $H_2$GCN Zhu et al. (2020) and FSGNN Maurya et al. (2022). Building on this, $H_2$GCN and TDGNN Wang & Derr (2021) aggregate higher-order neighborhood information across layers. Other methods focus on aggregation strategies, like OrderedGNN Song et al. (2023), which aligns the hierarchy of rooted trees with neuron order, adaptive channel mixing in ACM-GNN Luan et al. (2022), and gated kernels in GBK-GNN Du et al. (2022). While these methods enhance the internal structures of GNNs, models like Geom-GCN Pei et al. (2019) and WRGNN Suresh et al. (2021) refine the message-passing mechanisms for heterophilic graphs Qiu et al. (2024); Pan & Kang (2023), though they often involve complex, dataset-specific designs, limiting their generalizability. Additionally, spectral methods like GPR-GNN Chien et al. (2020), BernNet He et al. (2021), JocabiConv Wang & Zhang (2022), ALT Xu et al. (2023), and FAGCN Bo et al. (2021) use signed messages to capture high-frequency signals. However, these methods may suffer from the "negative times negative equals positive" effect, which is problematic for multi-class heterophilic graphs Liang et al. (2024). More recently, global information has been incorporated into models, including message-passing-based methods Li et al. (2022); Liang et al. (2024), and Graph Transformer-based methods Shi et al. (2021); Rampášek et al. (2022); Chen et al. (2023); Fu et al. (2024). However, these approaches incorporate global context through external mechanisms, such as global coefficient matrices or self-attention, without addressing the inherent limitations of the layer-by-layer structure in the message-passing mechanism.

Recent analyses have uncovered issues with several widely-used heterophilic graph datasets Platonov et al. (2023). For instance, datasets like *Squirrel* and *Chameleon* suffer from data leakage due to duplicate nodes, while smaller datasets such as *Cornell*, *Texas*, and *Wisconsin* face class imbalance and limited size (fewer than 1K nodes). Additionally, evaluations of heterophilic GNN models on new, larger datasets (10K-50K nodes) revealed that these advanced models often underperform, even lagging behind traditional GNNs like GCN Kipf & Welling (2016), GAT Veličković et al. (2018), and GraphSAGE Hamilton et al. (2017). Thus, there is a need for more profound analysis to understand the underlying causes and to develop solutions that are more robust and adaptable for real-world graph scenarios.

## 6 CONCLUSION

In this paper, we first conducted both theoretical and empirical analysis of the localized layer-by-layer nature of the message-passing mechanism. Then, we introduced a Global Message-passing Graph Neural Network (GloMP-GNN) for heterophilic graphs. By innovatively integrating a structure-based global propagation and a feature-augmented compensatory update module into the message-passing framework, GloMP-GNN effectively addresses the issue where messages from high-order but similar neighbor nodes are often weakened during propagation. We hope that our work will inspire further research in this direction. In future work, we plan to extend GloMP-GNN to other graph mining tasks, such as graph classification and downstream tasks like anomaly detection, as well as explore its potential in handling various types of graph data, such as heterogeneous graphs and hypergraphs.

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

## A APPENDIX

### A.1 PROOF OF THEROEMS

Firstly, we describe the formula derivation process of the $l$-layer GNN $\mathbf{X}^{(l)} = \prod_{i=1}^{l} \hat{\mathbf{A}}^{(l-i+1)} \mathbf{X}^{(0)} \mathbf{W}^{(i)}$. For simplicity, the $\sigma(\cdot)$ function (*i.e.*, $ReLU$) is omitted as mentioned before. Thus, for traditional GNNs,

$$\mathbf{X}^{(l)} = \hat{\mathbf{A}}^{(l)} \mathbf{X}^{(l-1)} \mathbf{W}^{(l)}$$
$$= \hat{\mathbf{A}}^{(l)} (\hat{\mathbf{A}}^{(l-1)} \mathbf{X}^{(l-2)} \mathbf{W}^{(l-1)}) \mathbf{W}^{(l)}$$
$$\cdots$$
$$= \prod_{i=1}^{l} \hat{\mathbf{A}}^{(l-i+1)} \mathbf{X}^{(0)} \prod_{i=1}^{l} \mathbf{W}^{(i)}.$$

Then, we prove Theorem 1.

**Proof of Theorem 1.**

Table 3: Statistics of the node classification datasets.

|  | **Actor** | **Roman** | **Amazon** | **Minesweeper** | **Tolokers** | **Cora** | **Citeseer** | **Pubmed** |
|---|---|---|---|---|---|---|---|---|
| #Nodes | 7,600 | 22,662 | 24,492 | 10,000 | 11,758 | 2,708 | 3,327 | 19,717 |
| #Edges | 26,659 | 32,927 | 93,050 | 39,402 | 519,000 | 5,278 | 4,552 | 44,324 |
| #Features | 931 | 300 | 300 | 7 | 10 | 1,433 | 3,703 | 500 |
| #Classes | 5 | 18 | 5 | 2 | 2 | 7 | 6 | 3 |
| $h_{\text{edge}}$ | 0.22 | 0.05 | 0.38 | 0.68 | 0.59 | 0.81 | 0.74 | 0.80 |
| LI | 0.00 | 0.11 | 0.04 | 0.00 | 0.01 | 0.59 | 0.45 | 0.41 |

**Proof 1** *For traditional GNNs,*

$$\mathbf{x}_i^{(l)} = \sigma\Big( \sum_{j \in \mathcal{N}(i)} c_{ij} \mathbf{W} \mathbf{x}_j^{(l-1)} \Big).$$

*Here, $c_{ij}$ is the weight coefficient of node $j$ to node $i$. And for heterophilic graphs, $c_{ij} \leq 1$ and don't tend to $1$.*

*The influence of a $k$-order neighbor $i_k$ to node $i_0$ in the path $\mathcal{P}$ is calculated as:*

$$C_{i_0 i_k}^{\mathcal{P}} = \prod_{j=0}^{k-1} c_{i_j i_{j+1}}.$$

*Thus, as $k$ grows, the influence intensity $C_{i_0 i_k}^{\mathcal{P}}$ becomes smaller.*

*Therefore, we conclude that:*

$$\lim_{k \to \infty} C_{i_0 i_k}^{\mathcal{P}} = 0.$$

## A.2 DATASETS DETAILS

In this part, we describe three homophilic datasets and five heterophilic datasets and the heterophily metric of these datasets. The statistics for these datasets are presented in Table 3.

**Homophilic Datasets**: *Cora*, *CiteSeer*, and *PubMed* Namata et al. (2012); Kipf & Welling (2016) are datasets derived from citation networks. In these datasets, nodes symbolize papers, while edges denote citations between them, and the label of a node indicates the academic subject of the paper. These datasets are categorized as homophilic datasets.

**Heterophilic Datasets**: *Actor* Pei et al. (2019) is a subgraph where nodes denote actors and edges signify co-occurrences on a Wikipedia page. Node features are Wikipedia page keywords, and the aim is to classify nodes into five categories based on their page content. *Roman-empire* Lhoest et al. (2021); Platonov et al. (2023) challenges GNNs with low homophily, sparse links, and long-distance dependencies. In the dataset, nodes represent words and are connected if they are consecutive or syntactically related in a sentence. *Amazon-ratings* Leskovec & Krevl (2014); Platonov et al. (2023) is based on the Amazon product co-purchasing network metadata dataset from SNAP[2] Datasets. In the dataset, nodes are products, and edges connect products that are frequently bought together. The task is to predict the average rating given to a product by reviewers. *Minesweeper* Platonov et al. (2023) is a dataset based on the Minesweeper game. The graph is a 100x100 grid where each node connects to up to eight neighbors. The challenge is to identify the 20% of nodes randomly set as "mines". *Tolokers* Platonov et al. (2023) comprises nodes symbolizing tolokers (workers) who have been a part of one of 13 chosen projects[3]. The dataset aims to predict which tolokers faced bans in a project.

**Heterophily Metric**: There are two metrics we used to evaluate the heterophily of datasets. In general, $h_{\text{edge}}$ has been the most often used metric, defined as:

---

[2]https://snap.stanford.edu/data/amazon-meta.html
[3]https://github.com/Toloka/ TolokerGraph

$$h_{\text{edge}} = \frac{|(u,v) \in \mathcal{E} : y_u = y_v|}{|\mathcal{E}|}, \tag{11}$$

where $y_u$ is the label of a node $u$ and $\mathcal{E}$ is the set of edges.

However, $h_{\text{edge}}$ is not suitable for datasets with unbalanced classes. Then, the LI metric is introduced to address these shortcomings. LI quantifies how much information a neighbor's label gives about the node's label, making it more versatile in various graph scenarios. It is defined by:

$$LI = \frac{I(y_u, y_v)}{H(y_u)}, \tag{12}$$

where $y_u$ and $y_v$ are random labels of $u$ and $v$ respectively, $H(y_u)$ represents the entropy of $y_u$, and $I(y_u, y_v)$ denotes the mutual information between $u$ and $v$.

### A.3 DESCRIPTION OF BASELINE

In this part, we describe 18 baselines that we used to compare with our models. And descriptions are listed as follows:

(1) Deep learning method:

- **ResNet** He et al. (2016) is a deep learning model utilizing residual connections for effective deep network training. In graphs, it views nodes as independent samples, while ignoring the graph structure.

(2) Classic GNN methods:

- **GCN** Kipf & Welling (2016) is a semi-supervised graph convolutional network model that learns node representations by aggregating information from neighbors.
- **GraphSAGE** Hamilton et al. (2017) is a framework for inductive representation learning on large graphs based on sampling algorithms.
- **GAT** Veličković et al. (2018) uses attention mechanisms to weigh neighbor contributions, allowing different neighbors to contribute differently to the node's new representation.
- **GATv2** Brody et al. (2022) improves upon GAT by introducing a more expressive and flexible dynamic attention mechanism.

(3) Selective information propagation methods:

- **H$_2$GCN** Zhu et al. (2020) integrates ego and neighbor-embedding separation, and higher-order neighborhoods, showing enhanced performance on heterophilic graphs.
- **GBK-GNN** Du et al. (2022) introduces a bi-kernel feature transformation, capturing both homophily and heterophily properties.
- **GCNII** Chen et al. (2020) is an extension of graph convolutional network with initial residual and identity mapping which can relieve the problem of over-smoothing.
- **FSGNN** Maurya et al. (2022) is a GNN model that uses "Soft-Selector" for adaptive feature choice and "Hop-Normalization" for improved node classification performance.
- **OrderedGNN** Song et al. (2023) is a GNN model that aligns the hierarchy of the rooted-tree of a central node with the ordered neurons in its node representation.

(4) Graph signal-based methods:

- **GPR-GNN** Chien et al. (2020) is a novel GNN architecture that adaptively learns Generalized PageRank (GPR) weights. It can effectively handle both homophily and heterophily and prevents feature over-smoothing.
- **FAGCN** Bo et al. (2021) utilizes a self-gating mechanism to adaptively integrate different signals in message passing, enhancing the adaptability of the model and addressing over-smoothing problems in various networks.

- **JacobiConv** Wang & Zhang (2022) is a spectral graph neural network approach that leverages Jacobi polynomial basis.
- **ALT-APPNP** Xu et al. (2023) is a structured-based method that decomposes a given graph and extracts complementary graph signals adaptively for node classification.

(5) Global information-based methods:

- **Graph Transformer (GT)** Shi et al. (2021) incorporates transformer architecture into GNNs. It uses self-attention mechanisms to capture global information in graph data.
- **GraphGPS** Rampášek et al. (2022) use self-attention mechanisms to capture global information while combining local message-passing and positional/structural encodings for improved scalability and performance.
- **GloGNN++** Li et al. (2022) introduce a global coefficient matrix to capture the correlations between nodes in each layer.
- **LRGNN** Liang et al. (2024) use a global label relationship matrix to replace the aggregation matrix by solving a robust low-rank matrix approximation problem.

## A.4 TIME ANALYSIS

We compare our model with GAT and Graph Transformer (GT) in terms of training time for 1000 epochs across the five largest datasets. As shown in Table 4, GloMP-GNN consistently outperforms GT in training time, while striking an effective balance between the efficiency of GAT and the broader information aggregation of GT. GAT achieves the shortest training times due to its local attention mechanism, which focuses on neighboring nodes, but at the expense of capturing global relationships. GT incorporates more complex transformations and global attention, leading to longer training times.

Table 4: Training Time Comparison for 1000 Epochs on the Five Largest Datasets.

| Model | Tolokers | Amazon | Minesweeper | Pubmed | Roman |
|---|---|---|---|---|---|
| GAT | 41s | 35s | 23s | 29s | 32s |
| GT | 68s | 58s | 32s | 46s | 53s |
| GloMP-GNN | 43s | 38s | 27s | 31s | 34s |

## A.5 FURTHER EXPERIMENTS ON OVER-SMOOTHING

In order to quantify the ability of GloMP-GNN to mitigate over-smoothing problem, we compute the Dirichlet Energy of 64 layers for GloMP-GNN after training. As shown in Table 5, GloMP-GNN exhibits significantly higher Dirichlet Energy on the Actor, Minesweeper, and Cora datasets compared to GCN and GCNII. This indicates that GloMP-GNN better preserves the diversity of node features, making it more resistant to the over-smoothing problem.

Table 5: Dirichlet Energy of 64 layers for GloMP-GNN after training. The higher energy indicates that it is less prone to over-smoothing.

| Model | Actor | Minesweeper | Cora |
|---|---|---|---|
| GCN | 0.1633 | 0.0007 | 0.0791 |
| GCNII | 0.3155 | 0.4312 | 0.1562 |
| GloMP-GNN | 0.7176 | 0.5936 | 0.2782 |

