# OpenReview forum: "Beyond Layers: A Global Message-Passing Mechanism for Heterophilic Graphs"
_ICLR.cc/2025/Conference — ICLR 2025 Conference Withdrawn Submission_

### Official Review · Reviewer_rYg8 · 2024-10-16

**Soundness:** 2
**Presentation:** 3
**Contribution:** 2
**Rating:** 3
**Confidence:** 4

**Summary:**

This paper points out the existence of low-order but dissimilar neighbor nodes, where the messages can be smoothed during the propagation. To better utilize their information, the authors propose two strategies: 1) structure-based global message-passing, and 2) feature-augmented compensation. The extensive experiments and ablation study show the effectiveness of the proposed scheme.

**Strengths:**

**S1.** The idea of receiving messages from distant nodes is reasonable. In addition, some figures (e.g., Fig 1) improve the readability of the manuscript

**S2.** The definitions and theorems are clearly defined

**S3.** Extensive experiments and ablation study show the effectiveness of the proposed method

**Weaknesses:**

**W1.** The novelty of the proposed method is quite vague. For example,
* Definition 1 (Global Intensity), definition 2 (Path Intensity), and definition 3 (k-order neighbors) are obvious as they are normalized based on the number of adjacent neighbors during propagation.
* The concept of Theorem 1 (over-smoothing) is already discussed and several studies propose to solve this problem [1, 2]. In addition, [3] proved that even GAT converges on an exponential rate. Lastly, the method of non-local message-passing free from node order is introduced in [4].
  * [1] Two sides of the same coin: Heterophily and oversmoothing in graph convolutional neural networks, ICDM '22
  * [2] Not too little, not too much: a theoretical analysis of graph (over) smoothing, NeurIPS '22
  * [3] Demystifying oversmoothing in attention-based graph neural networks, NeurIPS '24
  * [4] Non-local graph neural networks, TPAMI '22

Q1) Could you please explicitly state how the proposed definitions and theorem differ (if at all) from existing work?


**W2.** As mentioned by the authors, the proposed GEA (Global Edge Adaptation) extends the concept of GVN (Global Virtual Node) by applying the attention layer, which is the same as the GAT.

Q2) How does GEA specifically differs from or improves upon GAT? Could you please elaborate on the specific similarities and differences between these methods?


**W3.** The author suggests the integration of Gram matrix (which is the same as MLP) by measuring the similarity of initial node features and insist that this can increase the generalization ability. However, this contradicts to the bias-variance tradeoff. Generally, the prediction variance gets higher if it is trained without neighboring nodes (MLP). Even under the high heterophily, it has been shown [5] that utilizing the neighboring nodes can boost the performance by discovering the patterns of the adjacent nodes. From my view, the authors need to prove that the Gram matrix $M_G$ improves the generalization ability theoretically.
  * [5] Revisiting heterophily for graph neural networks, NeurIPS '22

Q3) How does this approach balance the bias-variance tradeoff and improves generalization, particularly in comparison to methods that utilize neighboring nodes? Could you provide a theoretical guarantee that Gram matrix improves the generalization ability of GNN?

**W4.** In equation 6, $\beta$ is given as trainable parameter, which can balance the influence of the retrieved edge coefficients. From my thinking, it can be biased towards 0 or 1 without specific constraint. The author needs to show the change of this value in the experiment section.

Q4) Could you please analyze how does this value evolves during training across different datasets or model configurations?


**W5.** The suggested feature-augmented compensatory update (Sec 3.2) looks like a simple combination of the previous methods as,
  * Eq. 7: Gram matrix (heterophily)
  * Eq. 8: GraphSAGE (homophily)
  * Eq. 9: GCN (homophily)

Q5) How does the combination of these methods improve the overall performance? It seems like Gram matrix (Eq. 7) contradicts to the others (Eq. 8 and 9). Can you please provide some explanation that this can improve the quality of the prediction?

**Questions:**

Please see the weaknesses

---

### Official Review · Reviewer_BStf · 2024-10-25

**Soundness:** 2
**Presentation:** 3
**Contribution:** 2
**Rating:** 3
**Confidence:** 4

**Summary:**

The authors propose a new graph neural network (GloMP-GNN), which can work well in both homophilic and heterophilic graphs.

The authors first analyze the limitations of layer-by-layer GNNs, indicating that the information is getting weaker as the propagation path is getting longer.

Based on this limitation, GloMP-GNN uses a global-level virtual node to mitigate this "vanishing information" phenomenon.

The authors experimentally verified the superiority of the proposed method compared to the several existing GNNs.

**Strengths:**

- S1. The paper is generally well-written and can be easily understood.
- S2. The provided figure provides an overview of the proposed method well.

**Weaknesses:**

- ***W1. Concerns on theories.*** My first concern is about the soundness of the theory. Theorem 1 indicates that the infinity-length path leads to zero information. In practice, since we do not stack a very large number of GNN layers, this result hardly highlights the limitation of the existing GNN methods. Rather, I think discussing diminishing ratio (i.e., information is shrinking with the exponential to the number of layers or quadratic to the number of layers, etc...) could be more adequate to pinpoint the limitation. Thus, I think the proposed theorem cannot well motivate the proposed method.

- ***W2. Concerns of novelty.*** My second concern is about the novelty of the proposed method. In my opinion, the proposed method is a reasonable combination of the existing method. (1) Global node is an idea widely used in graph transformers, (2) graph attention is also widely used in various GAT-based methods, and (3) embedding concatenation is proposed in H2GCN. Overall, while each component is adequate, I feel the proposed method somewhat lacks novelty.

- ***W3. Complexity.*** The authors highlight the limitation of the existing graph transformers (quadratic complexity w.r.t. number of nodes, lines 77-79), I think the proposed method shares this limitation, since the proposed method is also computing the gram matrix, which is $XX^{T}$. Did I correctly understand this limitation?

**Questions:**

Please see the weakness section. Thank you.

---

> ### Author Response · Authors · 2024-11-14
>
> Thank you for your thoughtful comments. We appreciate the opportunity to clarify our contributions and respond to each point individually.
>
> **W1.Concerns on theories**: Thank you for your suggestion. We appreciate the idea of examining the diminishing ratio, and we will consider incorporating this analysis in future work to further refine the theoretical foundations of our method.
>
> **W2.Concerns of novelty**: Our method is fundamentally motivated by the identified limitations of the layer-by-layer message-passing mechanism, rather than by simply building on existing ideas. Its novelty lies in addressing these limitations directly through the integration of global virtual nodes, adaptive edge adjustment, and multi-view feature updates, specifically designed to counteract message weakening in heterophilic graphs. This approach uniquely addresses the challenges in heterophilic settings and sets our method apart from previous work.
>
> **W3. Complexity**: The Gram matrix is computed only once as a preprocessing step, so it does not add quadratic complexity during training, keeping our method efficient.

---

### Official Review · Reviewer_MjQS · 2024-11-02

**Soundness:** 2
**Presentation:** 2
**Contribution:** 2
**Rating:** 3
**Confidence:** 5

**Summary:**

This paper analyzes the limitations of the message-passing mechanism in heterophilic graphs and introduces a novel approach called GloMP-GNN. GloMP-GNN incorporates global insights into both the message propagation and feature updating phases, using virtual edges and a multi-view feature updating mechanism to enhance node feature representation. Extensive experiments on eight datasets show that GloMP-GNN outperforms existing methods and mitigates the over-smoothing problem.

**Strengths:**

1. Improving virtual nodes using a Gram matrix is an interesting idea, and the proposed model appears to perform well on benchmarks.

**Weaknesses:**

1. **Method**

   - The approach proposed in the paper relies on the Gram matrix, which requires maintaining a matrix quadratic in relation to the number of nodes, making the model unscalable.

   - The multi-view proposed in the paper seems quite ad-hoc. Why were these three views chosen? It seems to be just a simple combination of GAT variant (the first view), APPNP (the second view), and GCN (the third view). Can this be understood as an ensemble model? This work incorporates too many previous methods as components, lacking sufficient motivation.

2. **Experiments**: The datasets used in the paper are not large in scale (with at most 20,000+ nodes), and do not demonstrate the scalability of the proposed method.

**Questions:**

1. In line 258 and line 283, the author mentioned "multi-head attention" twice, but why does the formula 3 author provided not include the multi-head mechanism?

---

> ### Author Response · Authors · 2024-11-14
>
> Thank you for your thoughtful comments. We appreciate the opportunity to clarify our design choices and their theoretical motivations.
>
> **1. Gram Matrix Scalability**: We would like to clarify that the Gram matrix is computed only once as a preprocessing step and does not participate in training. By doing so, we retain the benefits of capturing global structural information without incurring additional computational complexity during model updates, which supports the model’s scalability on larger graphs.
>
> **2. Choice of Multi-View Mechanism**: The three views were chosen not as an arbitrary combination but to provide a comprehensive representation of node features in heterophilic graphs. This setup allows messages weakened in one view to be effectively compensated by messages from the other views.
>
> **3.Experiments**: Thank you for the feedback. The datasets in our study effectively showcase the model's performance in challenging heterophilic graph scenarios. Our approach is designed to be efficient, with components such as the Gram matrix computed only once, ensuring scalability and efficiency.

---

### Official Review · Reviewer_FYk4 · 2024-11-02

**Soundness:** 2
**Presentation:** 3
**Contribution:** 1
**Rating:** 3
**Confidence:** 4

**Summary:**

This paper studies the node classification problem on graphs beyond heterophiliy. Its core idea is to add an extra virtual node.

**Strengths:**

S1. This paper is very clear and easy to understand.

S2. The experimental section includes many baseline methods and datasets.

**Weaknesses:**

W1. The main weakness of this paper is its novelty, which is very low, to the extent that many statements and ideas are well-known in the graph machine learning community. I would name a few:

W1.1 Theorem 1 claims that with the increase of propagation steps, the multiplication of (normalized) edge weights will be 0, which is a well-known fact that originated from the very early study regarding PageRank.

W1.2 Theorem 2 is a very obvious fact that, to be frank, cannot even be named a "Theorem."

W1.3 Eqs 8 and 9, which are the core methods of the section "FEATURE-AUGMENTED COMPENSATORY UPDATE," are just the same as the PageRank with different normalization (e.g., row normalization or symmetric normalization).

W1.4 The core idea of this paper, adding a global virtual node into the given graph to improve connectivity, has been thoroughly studied. For example, as early as one of the standard methods in the OGB benchmark [1], and some recent studies like [2] and [3].

W2. Some strong and recent baselines [4-7] are missing. After adding them back, the proposed method is not the best-performed one.

W3. This is a minor concern compared to the previous two. No theoretical analysis shows why the proposed method can work so effectively.

[1] Hu, Weihua, et al. "Open graph benchmark: Datasets for machine learning on graphs." Advances in neural information processing systems 33 (2020): 22118-22133.

[2] Fu, Dongqi, et al. "VCR-Graphormer: A Mini-batch Graph Transformer via Virtual Connections." The Twelfth International Conference on Learning Representations.

[3] Cai, Chen, et al. "On the connection between mpnn and graph transformer." International Conference on Machine Learning. PMLR, 2023.

[4] Jang, Hyosoon, et al. "Diffusion probabilistic models for structured node classification." Advances in Neural Information Processing Systems 36 (2024).

[5] Zheng, Amber Yijia, et al. "Graph Machine Learning through the Lens of Bilevel Optimization." International Conference on Artificial Intelligence and Statistics. PMLR, 2024.

[6] Luan, Sitao, et al. "Revisiting heterophily for graph neural networks." Advances in neural information processing systems 35 (2022): 1362-1375.

[7] Zhao, Kai, et al. "Graph neural convection-diffusion with heterophily." Proceedings of the Thirty-Second International Joint Conference on Artificial Intelligence. 2023.

**Questions:**

Please check the weaknesses I mentioned.

---

> ### Author Response · Authors · 2024-11-14
>
> Thank you for your detailed review and valuable feedback on our work. We appreciate the opportunity to address your concerns and clarify certain aspects of our methodology and theoretical contributions.
>
> **W1.1 Regarding Theorem 1 and its similarity to PageRank:**
>
> Thank you for this point. We would like to clarify that Theorem 1 is specifically designed for heterophilic graphs, and it does not claim that the multiplication of (normalized) edge weights will be zero in all types of graphs. PageRank also does not make this claim. In fact, in traditional homophilic graphs, the multiplication of (normalized) edge weights may remain non-zero.
> For reference, see examples such as in https://www.quora.com/Is-there-an-example-of-an-infinite-product-of-positive-numbers-less-than-1-that-converges-to-something-greater-than-0.
>
> **W1.2 Regarding Theorem 2:**
>
> Thank you for your suggestion. We will change the term to Proposition to more appropriately reflect its supporting role within our framework.
>
> **W1.3 Regarding Eqs. 8 and 9 being similar to PageRank:**
>
> While Eqs. 8 and 9 involve normalized aggregation, they are not identical to PageRank.
> PageRank is a diffusion-based method that typically aggregates information based on a fixed (static) adjacency matrix and focuses on the global random walk perspective. In contrast, our method uses adaptive aggregation based on local neighborhood information and multi-view feature updating, which is quite different from how PageRank operates.
>
> Specifically:
>
> Equation 8 performs a mean aggregation across neighbors, which is based on a node's local feature information, rather than the global transition probabilities used in PageRank.
>
> Equation 9 introduces normalized aggregation based on node degrees, which is a more refined, weighted form of aggregation, compared to PageRank's uniform normalization over neighbors.
>
> While both approaches involve normalization, the underlying mechanisms and objectives are different: our method is designed to augment node features by compensating for weak messages from different views, whereas PageRank is focused on random walk probabilities across the graph.
>
>
> **W1.4 Regarding the use of a global virtual node:**
>
> Thank you for this comment. While it is true that adding a global virtual node has been explored in prior works, the novelty of our approach lies in how we leverage the global virtual node specifically within the context of heterophilic graphs. Our key contribution is the integration of the global virtual node with adaptive edge weight adjustment (through Global Edge Adaptation) and feature-augmented compensatory updates, which are specifically designed to address message weakening issues unique to heterophilic settings. These enhancements enable our model to maintain robust connectivity and effectively propagate information across the graph, capturing valuable signals from distant, similar nodes that would otherwise be weakened.

---

### Note · Authors · 2024-11-19

I have read and agree with the venue's withdrawal policy on behalf of myself and my co-authors.